# In Vitro Infection Dynamics of Wuxiang Virus in Different Cell Lines

**DOI:** 10.3390/v14112383

**Published:** 2022-10-28

**Authors:** Xiaohui Yao, Qikai Yin, Danhe Hu, Shihong Fu, Weijia Zhang, Kai Nie, Fan Li, Songtao Xu, Ying He, Guodong Liang, Xiangdong Li, Huanyu Wang

**Affiliations:** 1Jiangsu Co-innovation Center for Prevention and Control of Important Animal Infectious Diseases and Zoonoses, College of Veterinary Medicine, Yangzhou University, Yangzhou 225009, China; 2Department of Arboviruses, NHC Key Laboratory of Biosafety, National Institute for Viral Disease Control and Prevention, State Key Laboratory for Infectious Disease Prevention and Control, Chinese Center for Disease Control and Prevention, Beijing 102206, China; 3Joint International Research Laboratory of Agriculture and Agri-Product Safety, The Ministry of Education of China, Yangzhou University, Yangzhou 225009, China

**Keywords:** Wuxiang virus, cell line, in vitro characterization, susceptibility, cytotoxicity

## Abstract

Wuxiang virus (WUXV) is a newly discovered Bunyavirales transmitted by sandflies. It is found to infect humans and chickens and can cause neurological symptoms and even death in mice. However, the susceptibility of different hosts and tissue-derived cells to this virus is unclear. In this study, we examined cells derived from murine (BHK-21, N2A), human (HEK-293T, SH-SY5Y), dog (MDCK), pig (PK-15), monkey (Vero), and chicken (DF1), which were inoculated with WUXV at 0.05 MOI, and monitored for monolayer cytopathic effect (CPE). Culture supernatants and cells were collected from 0 to 96 h post-infection, cell viability was determined by trypan blue staining, numbers of infectious virus particles were quantified using plaque tests, and viral nucleic acid contents were determined by RT-qPCR. The presence of WUXV N antigen in infected cells was detected by Western blotting (WB). In response to virus infection, BHK-21, MDCK, and PK-15 cells were characterized by a clear CPE, and we observed reductions in the proportion of viable cells after 96 h. By contrast, no significant CPEs were observed in the other cell lines. We detected increases in viral titers, viral nucleic acid content, and N antigen expression in BHK-21, MDCK, PK-15, HEK-293T, N2A, SH-SY5Y, and DF1 cells post-infection. Vero cells showed no CPE, and the findings for other tests were negative. In conclusion, we tested the susceptibility of different cell lines to WUXV, enhanced our current understanding of WUXV biology at the cellular level, and laid the foundations for further investigation of the underlying virus infection mechanisms.

## 1. Introduction

Wuxiang virus (WUXV), isolated in China in 2018, belongs to the order *Bunyavirales*, family *Phenuiviridae*, genus *Phlebovirus* [1]. The order *Bunyavirales* is the largest group of RNA viruses, comprising 14 families, including 496 species [2,3]. Similar to other Bunyaviruses, WUXV has a spherical virion of 80 to 120 nm in size enveloped within a lipid bilayer [4,5]. WUXV contains a negative-sense tripartite RNA genome comprising small (S), medium (M), and large (L) segments. The S segment encodes nucleocapsid protein (NP) and a nonstructural protein (NSP). The N protein necessary for RNA synthesis and virus propagation is the major structural element of both the viral ribonucleoprotein (RNP) and virion [6]. The M segment encodes a glycoprotein precursor (Gp), which is cleaved into mature N and C glycoproteins [7], whereas the L segment encodes an RNA-dependent RNA polymerase (RdRp) [8].

WUXV is an arbovirus isolated from sandflies, all samples of Phlebotomus were identified as *Phlebotomus chinensis* from the molecular biological level through the sequence analysis of mitochondrial cytochrome oxidase I gene [9,10]. Sandflies have a blood-sucking habit and are fond of human blood in addition to a variety of animal sources, including dog, chicken and sheep [11]. Except viruses, it can spread other pathogens such as Leishmania by sucking blood. While Shanxi, Gansu and other provinces in northern China are endemic areas of kala-azar. Because of the same vectors and similar infection seasons, patients infected by sandfly-transmitted virus and/or Leishmania and/or co-infections in kala-azar endemic areas are difficult to identify. In addition, due to the broad geographic distribution of sandflies, the diversity of pathogens carried and different symptoms of disease such as fever, bleeding, and even encephalitis, research on the public health threat of diseases caused by sandflies needs to be enhanced [12]. Up to now, our research group has isolated 64 WUXV viruses in Wuxiang county and Yangquan county, Shanxi province, central China. It has previously been reported that the virus causes serious neurological symptoms in mice, including a failure to consume milk, convulsions, the rigor of the limbs, lateral lying position, and negative turning reflex, and has even caused morbidity and death in newborn mice. A wide range of emerging and re-emerging infectious diseases has become a major threat to human health, approximately 75% of which are zoonoses [13]. Meanwhile, the positive rates of WUXV neutralizing antibodies detected in the serum samples of local healthy people and domestic chickens indicate that WUXV can infect humans and poultry [14], also a zoonotic disease. However, there have been few studies on WUXV either in vitro or in vivo.

According to the report of the International Committee on Taxonomy of Viruses (ICTV), the genus *Phlebovirus* contains numerous viruses, including the Rift Valley fever virus (RVFV), the sandfly-borne Toscana virus (TOSV), Sandfly fever Naples virus (SFNV), Sandfly fever Sicilian virus (SFSV), and Mukawa virus (MKWV) [2]. Vero cells have been used to amplify RVFV, SFNV, and SFSV [15,16,17], MKWV was isolated using Vero cells [18]; HEK-293T cells have been used to study the role of TOSV viral proteins in host immune escape [19], and BHK-21 and BSR-T7/5 cells have been used to establish reverse genetic systems for manipulating the genome of RVFV and studying its virulence mechanisms [20,21].

At present, it was reported that WUXV can replicate in BHK-21 cells and cause cytopathic effects [22]. To gain further insights on the proliferation characteristics of phleboviruses in vitro, we sought in this study to characterize susceptibility of different cell types (8 cell lines derived from 6 species) to WUXV experimentally, we examined the viral titer, nucleic acid load, WUXV N protein expression, toxicity and cytopathic changes in cells infected with WUXV. Based on our findings, we established a suitable in vitro cell model that could be used to study the infection and replication of WUXV and lay foundations for determining the pathogenesis of this virus.

## 2. Materials and Methods

### 2.1. Cell Lines

Baby hamster kidney (BHK-21) cells and African green monkey kidney (Vero) cells were grown in minimum essential medium (MEM) supplemented with 10% fetal bovine serum (FBS; Gibco, Grand Island, NY, USA). Madin–Daby canine kidney (MDCK) cells, porcine kidney cells (PK-15) cells, human embryonic kidney (HEK-293T) cells, mouse neuroblastoma N2a (N2A) cells, human neuroblastoma (SH-SY5Y) cells and chicken fibroblast (DF1) cells were cultured in Dulbecco’s modified Eagle’s medium (DMEM) supplemented with 10% FBS. Cell media were supplemented with 100 U/mL penicillin and 0.1 mg/mL streptomycin. All eight cell lines (more information were summarized in Table 1) were obtained from frozen stocks maintained in our laboratory. BHK-21, MDCK, PK-15, HEK-293T, N2A, SH-SY5Y, DF1 and Vero cells were cultured at 37 °C in a 5% CO_2_ incubator.

### 2.2. Sequence Determination and Analysis

The Wuxiang virus (WUXV) strains SXWX1813-2 (GenBank accession: S segment, MN454528.1; M segment, MN454527.1; L segment, MN454526.1) and SXYQ1872-1 (GenBank accession: S segment, no. MT786487.1, M segment, OP235904; L segment, OP235905) used in this study were obtained from stocks maintained in our laboratory and were originally isolated in 2018 from Wuxiang County, and Yangquan County, Shanxi Province, China, respectively. To determine the sequences of M segment and L segment of the SXYQ1872-1 genome, we extracted total RNA from each sample using a Viral RNA Mini Kit (QIAamp; Qiagen, Valencia, CA, USA), following the manufacturer’s instructions and used a Ready-To-Go kit (GE Healthcare, Little Chalfont, Buckinghamshire, UK) to synthesize complementary DNA (cDNA) from the extracted viral RNA for subsequent viral gene amplification. Viral gene amplification from isolates was performed using 25-µL PCR systems containing a cDNA template, GoTaq Green Master Mix 2 (Promega, Madison, WI, USA), and 10 pmol/L each of the forward and reverse WUXV M and L gene amplification primers (Appendix A). The outcomes of gene amplification were confirmed using 1% agarose gel electrophoresis before subsequent nucleotide sequence determinations based on Sanger dideoxy sequencing performed commercially by Sangon Biotech (Shanghai, China).

BLAST analyses were conducted using the viral gene nucleotide sequence. CLC Genomics Workbench software (Qiagen, #832021) was used for sequence splicing and quality analyses, and MegAlign software (DNAStar, Madison, WI, USA) was used for the analysis of nucleotide and amino acid sequence similarities.

### 2.3. Viral Titer Determination

Both viral isolates were propagated in BHK-21 cells, with the respective viral titers being determined using a standard plaque assay. Briefly, a series of 10-fold dilutions of the virus was added to the wells of six-well culture plates. Following incubation, 4 mL of 1.1% methylcellulose-MEM semi-solid medium containing 2% FBS was added to each well. After culturing for several days, we observed clear plaque development. The cells were stained with crystal violet solution, and we calculated the plaque-forming units and viral titers [23].

### 2.4. Virus Growth Kinetics

Cells were infected with SXWX1813-2 and SXYQ1872-1, as previously described, and both cell and supernatant samples were collected at 0, 12, 24, 36, 48, 72 and 96 h post-infection (hpi). To determine a baseline of virus quantity, cells were additionally frozen immediately after 1 h adsorption followed by washing steps. After cell culture supernatants and whole cells had been harvested via freeze–thaw cycles, they were then stored at −80 °C. The virus was titrated using a standard plaque assay.

### 2.5. Quantitative Real-Time Polymerase Chain Reaction

To quantify the amounts of the viral genomic RNA (gRNA) harvested at different time points during incubation, RNA was extracted using a QIAamp Viral RNA Mini Kit (QIAGEN, Germantown, MD, USA) according to the manufacturer’s instructions. A quantitative real-time polymerase chain reaction (qRT-PCR) was conducted using an AgPath-ID^TM^ One-Step RT-PCR Kit (Applied Biosystems, Foster City, CA, USA). The primers, probes, and RNA standards were prepared, as previously described [24].

### 2.6. Cell Cytopathic Changes and Cell Viability

Cells (1 × 10^6^ cells per well) were seeded in six-well plates and left overnight before performing an infection assay. The cells were infected with SXWX1813-2 and SXYQ1872-1 at a multiplicity of infection (MOI) of 0.05 in MEM. After leaving for 1 h to facilitate virus adsorption, the monolayers were washed with sterile phosphate-buffered saline, followed by the addition of 2 mL of the corresponding cell culture medium containing 2% FBS, penicillin, and streptomycin. For each cell line and each virus strain, infection was performed at two separate time points. Following infection, cell morphology was monitored, and cytopathic effects (CPEs) were examined at 0, 48, and 96 h using a Nikon, ECLIPSE Ts2 inverted light microscope (Nikon, Tokyo, Japan). Cell viability was determined using a 1:1 dilution of 0.4% trypan blue stain solution (Solarbio, Beijing, China), and the cells were counted under the inverted light microscope using a cell counter (EASYBIO, Beijing, China).

### 2.7. Western Blotting

Cells were cultured in six-well plates, infected with SXWX1813-2 at 0.05 MOI, and left to stand for 1 h to facilitate adsorption. The cells were then collected at 0, 1-, 2-, 3- and 4-days post-infection (dpi), and subsequently lysed using RIPA buffer containing phenylmethylsulfonyl fluoride (Beyotime, Shanghai, China). Total cell protein was extracted, and protein concentrations were determined using a BCA protein assay kit (Beyotime Biotechnology, Jiangsu, China). Aliquots (60 µg) of the cell lysates were separated on 10% SDS–polyacrylamide gels and transferred to nitrocellulose membranes for Western blotting. The membranes were initially blocked with Tris-buffered Tween 20 (TBST) containing 5% non-fat dry milk for 2 h at room temperature (RT), after which the membranes were incubated overnight with the following primary antibodies at 4 °C: GAPDH mouse monoclonal antibody (1:10,000; AC002, ABclonal Biotechnology, Wuhan, China) and a polyclonal antibody against WUXV N protein (1:1000) were maintained in our laboratory. The membranes were then incubated with goat anti-mouse secondary antibodies (1:10,000) for 1 h at RT. Finally, the membranes were incubated with an enhanced chemiluminescent reagent and developed using an automatic chemiluminescence image analysis system (Amersham Imager 600; GE Healthcare, Little Chalfont, Buckinghamshire, UK). Densitometric values of protein bands were quantified using Image J software.

## 3. Results

### 3.1. Similarity Analysis of the Open Reading Frames of WUXV SXWX1813-2 and SXYQ1872-1

M and L segments of the WUXV strain SXYQ1872-1 were deposited into GenBank with accession numbers of OP235904 and OP235905. The nucleotide length of the open reading frame (ORF) region of the two virus strains was proved to be identical. The nucleotide length of RdRp encoded by the L gene is 6273 bp, with nucleotide and amino acid identities of 97.7% and 99.4%, respectively; the nucleotide length of the Gp glycoprotein encoded by the M gene is 4089 bp, with nucleotide and amino acid identities of 96.7% and 97.4%, respectively; and the nucleotide lengths of the NS and N genes encoded by the S segment are 783 bp and 741 bp, and the nucleotide and amino acid identities of both are 100% (Table 2). In addition, we identified twenty-seven and nine different amino acids in the ORF regions of M and L, respectively (Appendix A).

### 3.2. WUXV Induces Differential Cytopathic Changes and Cytotoxicity in Cell Lines

Among the eight selected cell lines examined in this study, we found that only BHK-21, MDCK, and PK-15 showed a clear CPE in SXWX1813-2- and SXYQ1872-1-infected cells and with the prolongation of infection time, CPE became more and more obvious, during which the cells gradually shrank and exfoliate (Figure 1). It was found that the degree and timing of the CPE caused by the two strains were the same. Among the three cell lines, BHK-21 and MDCK cells were characterized by a significant CPE at 48 hpi, with almost 80% of the cells having shrunk and fallen off at 96 hpi. At 96 hpi, the mock-infected BHK-21 and MDCK cultures showed evidence of substantial overgrowth with sloughing and piling, respectively, of the cell monolayers. The number of collected cells and the proportion of viable infected cells, as measured by trypan blue exclusion, were largely consistent with the cytolysis observed microscopically at early time points (Figure 2A,B). The CPE of PK-15 cells at 48 h was not as obvious as that observed in the BHK-21 and MDCK lines, but these cells also showed aggregation compared with the control group.

### 3.3. Effect of the Virus on Cell Viability

Our examination of the effect of WUXV on the viability of the aforementioned four cell lines revealed that WUXV showed higher cytotoxicity to BHK-21, MDCK, and PK-15 cells than to those of HEK-293T cells. Moreover, we found that the number of viable cells at 96 h was lower in infected cells than in uninfected cells (Figure 2).

### 3.4. Cells Lines Support WUXV Replication and Produce Infectious Virus

In addition to the aforementioned three cell lines characterized by a CPE, the two viruses (SXWX1813-2 and SXYQ1872-1) were also able to replicate and proliferate in four of the other five cell lines examined, namely, HEK-293T, N2A, SH-SY5Y, and DF1 cells, with only Vero cells appearing to show resistance to the virus (Table 1). However, we detected no significant differences among the seven susceptible cell lines with respect to the replication speed of the two viruses (Figure 3). Virus titers tended to increase with infection time, with those in MDCK, PK-15, and SH-SY5Y cells peaking at 24 h, those in BHK-21, HEK-293T, and DF1 cells peaking at 36 h, and those in N2A cells reaching a maximum at 48 h. With the exception of the N2A cell line, in which the virus titer declined at a later stage of infection due to cell death, having reached peak values, the virus titers of the other six cell lines remained stable until 96 hpi. These trends in viral titer were found to be consistent with our measurements of viral RNA contents at different time points using a previously established RT-qPCR detection method.

### 3.5. Western Blot Analysis of N Protein Expression in Different Cell Lines

Having exposed the test cells to the virus, cells were harvested after 1 h of viral adsorption, and at this timepoint, we detected virtually no expression of WUXV N protein in any of the assessed cells. However, among the eight cell lines, N protein expression was observed to be highest in BHK-21 and HEK-293T cells after virus infection and gradually increased over the subsequent 4 days. By contrast, N protein expression in the PK-15, SH-SY5Y, and DF1 cells peaked on the 1st day after infection, and thereafter underwent a gradual decline. Compared with the other cell lines, post-infection N protein expression was relatively lower in SH-SY5Y and DF1 cells. Whereas in MDCK cells, levels of N protein increased on day 1 after infection and thereafter were maintained at a relatively stable level. In NA2 cells, levels of N protein expression peaked 2 days after virus infection and subsequently underwent a decline before eventually stabilizing by day 4 (Figure 4).

## 4. Discussion

WUXV is a newly identified Bunia virus isolated from sandflies, and since its discovery, there has been very little research on the cellular infectivity of this virus. Among what limited research has been conducted, WUXV has only been reported to cause a cytopathic effect (CPE) in the BHK-21 cells [1], whereas its infectivity toward the cells of other species currently remains undetermined. Consequently, to enhance our limited understanding of the biological properties of WUXV at the cellular level, we evaluated the susceptibility of eight cell lines derived from a range of different tissues and species to WUXV.

The WUXV strains SXWX1813-2 and SXYQ1872-1 used in this study were isolated from two different regions of China, namely, Wuxiang County and Yangquan City, respectively, both of which are located in Shanxi Province. For this study, N gene was used for genetic evolutionary analysis to divide WUXV into type I and type II, with SXYQ1872-1 and SXWX1813-2 as the representative strains for type I and II, respectively [14]. These two strains were sequenced to obtain the complete sequences of all ORF regions. The sequencing results indicated that the two viruses differed little at the sequence level and no significant difference was observed between the two viruses with respect to the production of a CPE, virus titer, or nucleic acids in the infected cells. Therefore, SXWX1813-2 strain was used in the following experiments. 

Given the characteristics of the wide host range of arboviruses, we selected host cell lines derived from six different animal taxa, namely human, mouse, dog, chicken, pig, and monkey. Previous studies have shown that following Rift Valley fever virus (RVFV) infection in mice, the virus can be detected in the kidneys [25]. The monkey kidneys infected with RVFV are characterized by histopathological lesions [26]. Accordingly, in the present study, we selected four kidney-derived cell lines, namely, Vero, HEK-293T, MDCK, and PK-15, to examine the effects of viral infection. Other studies have shown that WUXV infection in mice can cause neurological symptoms [14]. Toscana virus (TOSV), a virus in the same genus as WUXV, can cause meningitis [27,28]. Two viremic piglets had a lymphoplasmacytic encephalitis with glial nodules after affected with RVFV [29]. Therefore, in present study, we also selected the two neuroblastoma cell lines SH-SY5Y and N2A cells.

The information on the aforementioned cell lines and the susceptibility of these cells to WUXV infection were summarized based on the following three aspects. Firstly, we determined the viral titer by using plaque assays, which provide estimates of live infectious virus particles. Secondly, to assess the nucleic acid production of the virus, we determined viral RNA levels at different time points during infection based on RT-qPCR analyses of a conserved region of the WUXV S gene. Thirdly, as the main structural element of viral ribonucleoprotein (RNP) and viral particles, the N protein is directly involved in RNA synthesis and viral reproduction [30]. Consequently, we performed Western blot analyses of the viral N protein to determine whether the selected host cells could support the expression of viral proteins. Based on our observations, we were able to divide the eight cell lines into three categories. The first of these, comprising the cell lines BHK-21, MDCK, and PK-15, was defined by four characteristics, namely, an obvious CPE, and increases in viral titer, viral nucleic acids, and viral proteins. The second category, into which we grouped PK-15, N2A, SH-SY5Y, and DF1 cells, was also characterized by increases in viral titer, nucleic acids, and proteins, although no evident CPE. The third category, in which we placed Vero cells, yielded negative results for all four of the assessed indicators.

Our observations revealed that among the seven WUXV-sensitive cell lines, three cell lines (MDCK, PK-15, and BHK-21) were kidney-derived and characterized by an obvious CPE and heightened cytotoxicity. In this regard, it has previously been observed that in a mouse model of severe fever with thrombocytopenia syndrome virus (SFTSV), which is classified in the same family as WUXV. The SFTSV-infected rodents were characterized by pathological damage to the renal capsule and showed symptoms of renal dysfunction [31]. Furthermore, in lambs infected with RVFV, the kidney lesion progresses were characterized by cell swelling and karyolysis, and multifocal acute renal tubular injury (nephrosis) in adult sheep with RVF was confirmed [32,33,34,35,36]. Accordingly, the findings of these studies highlight the need to focus on pathological changes in the kidneys and related urinary and reproductive systems, in further clinical sampling or animal experiments. However, we also established that HEK-293T cells, which are also kidney-derived, although highly susceptible to WUXV, do not show any apparent CPE. As we know, Vero cells can support the replication and proliferation of RVFV, TOSV and SFSV [15,16,17], which belong to the same family as WUXV. However, our findings suggest that Vero does not support the replication of WUXV, which is what we want to know, and we need to study it further.

WUXV infection in mice has been established to cause neurological symptoms [14], which tends to indicate that the virus may invade the nervous system. In the present study, we observed that WUXV could replicate in N2A and SH-SY5Y neuroblastoma cells. In N2A cells, we detected increases in viral nucleic acids, titer, and protein with a prolongation of the infection time. In SH-SY5Y cells, the virus replicated and proliferated during the early stage of infection, although we detected a reduction in the amounts of viral N protein on the second day, with levels gradually declining over the following 4 days of observation. However, we observed no clear CPE in either N2A or SH-SY5Y cells, and gerbils infected with the same genus of virus RVFV developed fatal encephalitis without obvious neuropathy [37], suggesting that although the virus may invade the nervous system, the pattern of injury is unclear, or that it may cause only mild, undetectable clinical symptoms. These findings also emphasize the need to further focus on the epidemiology of WUXV and enhance efforts to monitor cases of unknown fever or unknown encephalitis in areas in which vectoring sandflies are found.

In this study, we compared the susceptibility of human and animal cell lines derived from different tissues to WUXV, and our findings in this regard will provide an important reference for the future use of cell lines in vitro studies. Notably, we established that WUXV could potentially infect and be harmful to a range of different species. High viral loads of WUXV in kidney- and brain-derived cell lines provide evidence to indicate the possible target tissues for WUXV infection in vivo, and consistently significant CPE in kidney-derived cells suggests possible target organs for WUXV attack.

According to our results, BHK-21 was the most sensitive cell line among the 8 cell lines, with obvious CPE after virus infection and the highest virus titer, nucleic acid and protein content. Accordingly, BHK-21 could be recommended for virus isolation and culture, whereas N2A, HEK-293T, DF1, and BHK-21 cells could be used to study the pathogenesis of WUXV.

As a relatively newly isolated virus, there has to date been relatively little relevant research on WUXV, and the present study is the first in which the replication and proliferation of this virus in different cell lines has been examined. However, the study does have certain limitations, and as such, should be considered a preliminary investigation. Notably, although we have identified a range of viral characteristics and cellular responses, we have not sought to elucidate the underlying mechanisms. In further studies, we intend to investigate differential transcriptional responses in different cell lines to precisely characterize the intracellular events associated with WUXV infection, and to examine the mechanisms associated with the differential behavior of the virus in different cell lines. Nevertheless, despite the limited scope of the present study, our results lay the foundations for further study of the intracellular infection mechanism of WUXV, which will contribute to future research on WUXV biology and in the assessment of host factors and disease mechanisms in naturally infected hosts.

## 5. Conclusions

In conclusion, among the cell lines assessed in this study, BHK-21, MDCK, PK-15, HEK-293T, N2A, SH-SY5Y, and DF1 cells were established to be susceptible to WUXV infection, whereas Vero cells appear to be resistant to infection. Of the susceptible lines, BHK-21, MDCK, and PK-15 showed clear post-infection cytopathic effects, which grew more pronounced with an increase in infection time. WUXV was found to be cytotoxic to all three of these cell lines, thereby reducing cellular activity following infection. Our findings will provide an important reference for the use of cell lines in ex vivo studies and highlight the need for careful consideration based on their host and tissue origin, as well as the necessity of examining cytopathic effects and other implicated characteristics.

## Figures and Tables

**Figure 1 viruses-14-02383-f001:**
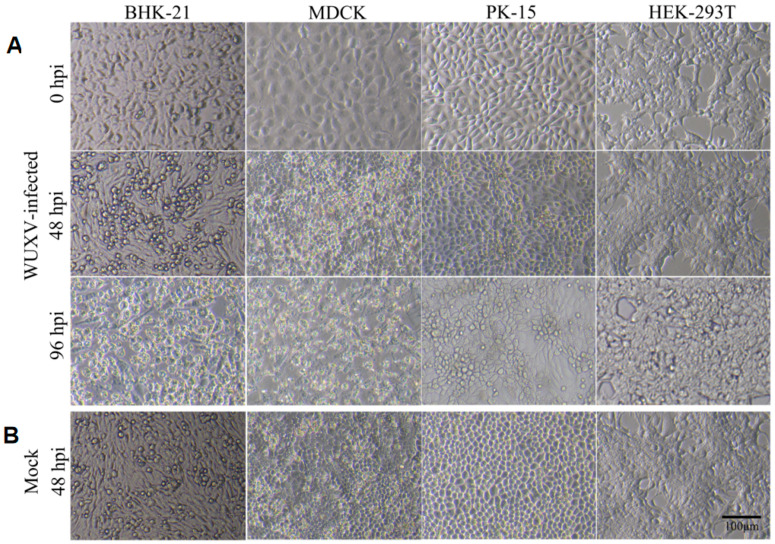
Cytopathic changes observed in cell lines at 0, 48, and 96 hpi. Cells were (**A**) infected with WUXV SXWX1813-2 or (**B**) mock infected (48 hpi) at a multiplicity of infection (MOI) of 0.05. Morphological changes were observed using brightfield microcopy; Scale Bar = 100 μm.

**Figure 2 viruses-14-02383-f002:**
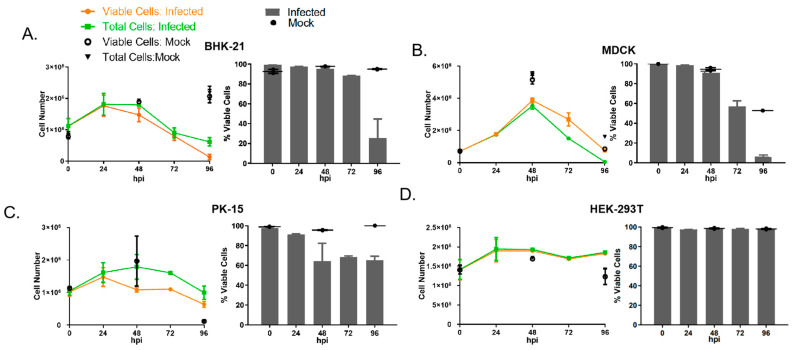
Cell numbers and viability following infection with WUXV SXWX1813-2. (**A**) BHK-21, (**B**) MDCK, (**C**) PK-15 and (**D**) HEK-293T cells were infected with WUXV SXWX1813-2 at a multiplicity (MOI) of 0.05 and collected at 0, 24, 48, 72, and 96 hpi; mock-infected samples were collected at 0, 48, and 96 hpi. Left panels show the absolute numbers of total cells and viable cells per well at each timepoint based on trypan blue exclusion. Right panels show the percent viability at each timepoint. Bars represent arithmetic mean ± SD of two biological replicates.

**Figure 3 viruses-14-02383-f003:**
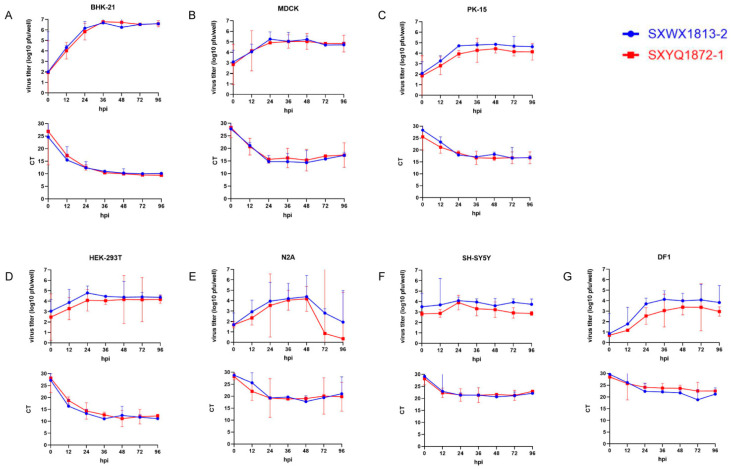
Viral and genomic replication of WUXV SXWX1813-2 and SXYQ1872-1 in different cell lines. (**A**) BHK-21, (**B**) MDCK, (**C**) PK-15, (**D**) HEK-293T, (**E**) N2A, (**F**) SH-SY5Y and (**G**) DF1 cell monolayers were infected with WUXV SXWX1813-2 and SXYQ1872-1 at a multiplicity (MOI) of 0.05, respectively. Cell and supernatant samples were collected at 0, 12, 24, 36, 48, 72, and 96 hpi. Upper panels show the viral titers that quantified by standard plaque assay and express as pfu/well for SXWX1813-2 (blue) and SXYQ1872-1 (red) fractions. Lower panels show the viral nucleic acid level by RT-qPCR and express as CT values for SXWX1813-2 (blue) and SXYQ1872-1 (red) fractions.

**Figure 4 viruses-14-02383-f004:**
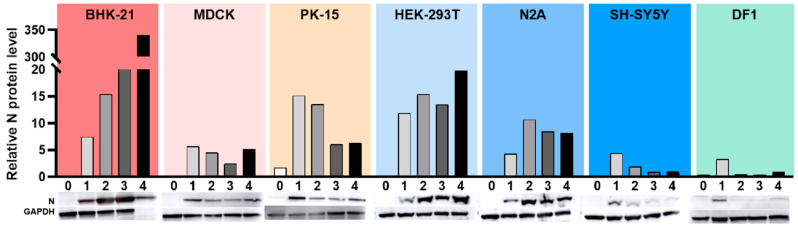
Detection of WUXV N protein in cells infected with SXWX1813-2. Cells infected with SXWX1813-2 at a multiplicity of infection (MOI) of 0.05 for 0d, 1d, 2d, 3d, 4d were lysed and analyzed by Western blot using the anti-N polyclonal antibody of WUXV. Protein expression was normalized to GAPDH. Densitometric values of protein bands were quantified by the Image J. Data were analyzed using GraphPad Prism™ software 8.0.2 (GraphPad Software, Inc., San Diego CA, USA).

**Table 1 viruses-14-02383-t001:** Cell lines tested for WUXV infection susceptibility.

Cell Line	Source	Tissue	Replication	CPE	Peak Titer (pfu/mL)		Time of Peak Titer (hpi)	
					SXWX1813-2	SXYQ1872-1	SXWX1813-2	SXYQ1872-1
BHK-21	hamster	kidney	+	+	4.7 × 10^6^	6.0 × 10^6^	36	36
MDCK	dog	kidney	+	+	1.8 × 10^5^	1.1 × 10^5^	24	48
PK-15	pig	kidney	+	+	7.1 × 10^4^	2.7 × 10^4^	48	48
HEK-293T	human	embryonic kidney	+	−	2.9 × 10^5^	1.6 × 10^5^	36	48
N2A	mouse	neuroblastoma	+	−	3.5 × 10^4^	1.5 × 10^4^	48	48
SH-SY5Y	human	marrow	+	−	1.2 × 10^4^	2.8 × 10^3^	24	24
DF1	chicken	embryo	+	−	1.4 × 10^4^	2.5 × 10^3^	36	48
Vero	monkey	kidney	−	−	/	/	/	/

Note: WUXV, Wuxiang virus; BHK-21, baby hamster kidney cells; MDCK, Madin-Darby canine kidney cells; PK-15, porcine kidney cells; HEK 293T, human embryonic kidney HEK 293T cells; N2A, mouse neuroblastoma N2a cells; SH-SY5Y; human neuroblastoma cells; DF1, chicken fibroblast cells; Vero, African green monkey kidney cells; hpi, hour post infection. −, non-permissiveness to WUXV; +, permissiveness to WUXV.

**Table 2 viruses-14-02383-t002:** Identities analysis of the ORF of WUXV SXWX1813-2 and SXYQ1872-1.

	S Segment						M Segment			L Segment	
**Virus Strains**	NS			N			GP			RdRp	
	nt (%)	aa (%)		nt (%)	aa (%)		nt (%)	aa (%)		nt (%)	aa (%)
SXWX1813-2	783	261		741	247		4089	1363		6273	2091
SXYQ1872-1	783 (100)	261 (100)		741 (100)	247 (100)		4089 (96.7)	1363 (97.4)		6273 (97.7)	2091 (99.4)

## Data Availability

All the data generated during the current study are included in the manuscript and/or the Appendix A. Additional data related to this article may be requested from the corresponding authors.

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
