# Peer review of "In Vitro Infection Dynamics of Wuxiang Virus in Different Cell Lines"

_viruses, 2022, doi:10.3390/v14112383_

Round 1

Reviewer 1 Report

With the work entitled “In Vitro Infection Dynamics of Wuxiang virus in different Cell Lines” the authors described the characterization of the infection by the new WUXV virus in different cell types. These data are important for the initial biological knowledge of the physiological dynamics of the virus and its pathogenicity, opening perspectives on possible drug developments used in the future if necessary. The professional language in English is consistent, as are the results presented through graphs and tables. The hypothesis discussed is in agreement with the data presented and with the literature provided at the end of the manuscript.

I have some suggestions of modifications to be made on text, tables and figure presentations (explained below, at the general comments for the author.

1. The introduction (and also in the Discussion) lacks information about virus vectors. Which Phlebotomus species transmit? The authors could enrich with this information, given that it is a pathogen transmitted by an insect that performs hematophagy in different animals.

2. In addition to this point mentioned above, the authors could succinctly correlate this zoonotic emergence with the existing co-infections with other parasites also transmitted by different species of Phlebotomus, such as protozoa of the genus Leishmania, in which they could worsen cellular pathogenicity. Of course, there are several current references on this topic.

3. Is there any information about the immune response against this virus, such as the synthesis of inflammatory mediators?

4. On page 26 the cell name is HEK-293T.

5. On page 77 the sentence is missing a full stop.

6. In item 2.5 in Material and Methods, which gene and primers were used to normalize (endogenous) the qRT-PCR reaction? In the supplementary table has not described. If one was actually used, I believe there should not be one that can be compatible with all the different cells used.

7. In item 2.7, the authors described that they used mouse anti-GAPDH as a normalizer. But can this antibody recognize GAPDH from humans, chicken, canines, monkeys and pigs, as shown in Figure 4?

8. On page 177 the sentence is missing a full stop.

9. In figure 1, where are the images of the other cells, as they seem to be able to replicate the viruses?

10. Figure 2 has some problems. First, the authors should better elucidate whether the 72h and 96h points after infection in PK-15 cells are statistically significant. Second, in the legend of this figure 2 the cell orders are reversed and wrong when compared to the graphs. Please correct.

11. Figure 3: Were cells C6/36 used, as they are in the legend?

12. The legend of figure 3 has the order of the cells also changed, as in figure 2. Please correct it.

13. On page 234 the sentence after “respectively” is missing a period.

14. On page 177 it is plaque, not plaqeu.

15. Figure 4: Some images of the bands do not match the images of the gels revealed in the supplementary material. I believe they are switched. Example: the image of the N protein of the MDCK cell 293T correspond to HEK cells bands, as well as the endogenous one. The N2A image has the same problem. Please review and correct the images.

16. Why in the supplementary images of the WB there are gels revealed with both antibodies at same membrane and others presented in 2 separate gels? Dis authors strip the membranes?

17. Page 258: Bunyavirus.

18. Page 270: End point after “ORF regions”.

19. Page 289: End point after “viral particles”.

20. Page 323: End point after “Infection time”.

21. Did the authors consider using a Leukocyte cell line to add information to the data presented, given that they also deal with the involvement of the immune system in response to the virus?

22. Although VERO cells appear resistant and do not form plaques, did the authors verify if the cell was able to replicate the virus through qRT-PCR and/or anti-N-protein WB?

Author Response

Response to reviewer 1 comments

With the work entitled “In Vitro Infection Dynamics of Wuxiang virus in different Cell Lines” the authors described the characterization of the infection by the new WUXV virus in different cell types. These data are important for the initial biological knowledge of the physiological dynamics of the virus and its pathogenicity, opening perspectives on possible drug developments used in the future if necessary. The professional language in English is consistent, as are the results presented through graphs and tables. The hypothesis discussed is in agreement with the data presented and with the literature provided at the end of the manuscript.

I have some suggestions of modifications to be made on text, tables and figure presentations (explained below, at the general comments for the author).

We really appreciate you for your rigorous and scientific attitude. Your suggestions are really valuable and helpful for revising and improving our paper. According to your suggestions, we have made the following revisions on this manuscript: 

Point 1: The introduction (and also in the Discussion) lacks information about virus vectors. Which Phlebotomus species transmit? The authors could enrich with this information, given that it is a pathogen transmitted by an insect that performs hematophagy in different animals.

Response 1: Thank you for your suggestion. The samples of Phlebotomus were identified as Phlebotomus chinensis from the molecular biological level through the sequence analysis of mitochondrial cytochrome oxidase I gene (Reference 9-10), and the WUXV we used in this paper were isolated from these phlebotomine samples. We have added the relevant information and references according to your suggestion in line 55-67 in the revised version.  

Point 2: In addition to this point mentioned above, the authors could succinctly correlate this zoonotic emergence with the existing co-infections with other parasites also transmitted by different species of Phlebotomus, such as protozoa of the genus Leishmania, in which they could worsen cellular pathogenicity. Of course, there are several current references on this topic.

Response 2: Thank you for your suggestion. We have discussed it in line 55-67 in the revised version.

Point 3: Is there any information about the immune response against this virus, such as the synthesis of inflammatory mediators? Response 3: Thank you for your question. There is no research on the immune aspect of the virus in our study, however it would be interesting to explore it, and we would research on it in the future. Point 4: On page 26 the cell name is HEK-293T. 

Response 4: We thank the reviewer for pointing this out. We have modified the cell name in our revised MS.

 Point 5: On page 77 the sentence is missing a full stop.

Response 5: We thank the reviewer for pointing this out. We have modified in our revised MS.

Point 6: In item 2.5 in Material and Methods, which gene and primers were used to normalize (endogenous) the qRT-PCR reaction? In the supplementary table has not described. If one was actually used, I believe there should not be one that can be compatible with all the different cells used.

Response 6: Thank you for your comments. We designed primers for the conserved region on WUXV S segment, and detected the virus in different cell lines by qRT-PCR, so as to show the replication of the virus at the nucleic acid level. Because the qRT-PCR detection method has been established (Reference 24). We have added the information of primers and probe to the supplementary table 4 according to your suggestion.

The primers and the probe of this method can detect viral nucleic acids in all sensitive cell lines.

Point 7: In item 2.7, the authors described that they used mouse anti-GAPDH as a normalizer. But can this antibody recognize GAPDH from humans, chicken, canines, monkeys and pigs, as shown in Figure 4?

Response 7: Thank you for your comments. The mouse anti-GAPDH used in our research was purchased from Abclonal(Catalog NO.: AC002) company, and the user-verified application in the instructions showed that the monoclonal antibody could be used for WB reaction of human, monkey, pig, chicken and dog. At the same time, our experimental results also showed that the antibody can recognize the GAPDH proteins in these species. We have added the brand of this antibody in our revised MS in line 180.

Point 8: On page 177 the sentence is missing a full stop.

Response 8: We thank the reviewer for pointing this out. We have modified in our revised MS.

 Point 9: In figure 1, where are the images of the other cells, as they seem to be able to replicate the viruses? Response 9: Thank you for your question. Among the seven infectious cell lines, CPE was observed in only three cell lines (BHK-21, MDCK, PK-15). In the other cell lines without CPE, qRT-PCR, plaques experiment and N protein expression were all positive. Similar virus had been isolated previously in our lab, Hedi Virus, which can replicate and proliferate in BHK-21 without CPE (Reference 9). This phenomenon in WUXV is interesting, and it' s one of the points we' re going to research next.  Point 10: Figure 2 has some problems. First, the authors should better elucidate whether the 72h and 96h points after infection in PK-15 cells are statistically significant. Second, in the legend of this figure 2 the cell orders are reversed and wrong when compared to the graphs. Please correct. 

Response 10: Thank you for your comments. First, there was no significant difference between 72hpi and 96hpi in PK-15 cells by T test analysis. Second, we are extremely grateful to reviewer for pointing out this mistake. We have modified in the legend of figure 2.

  Point 11: Figure 3: Were cells C6/36 used, as they are in the legend?

Response 11: Thanks for your question. We did not perform relevant experiments in C6/36 cells.

 Point 12: The legend of figure 3 has the order of the cells also changed, as in figure 2. Please correct it. 

Response 12: We thank the reviewer for pointing this out. We have modified in the legend of figure 3.

Point 13: On page 234 the sentence after “respectively” is missing a period.

Response 13: We thank the reviewer for pointing this out. We have modified in our revised MS.

Point 14: On page 177 it is plaque, not plaqeu.

Response 14: We thank the reviewer for pointing this out. We have modified in our revised MS.

Point 15: Figure 4: Some images of the bands do not match the images of the gels revealed in the supplementary material. I believe they are switched. Example: the image of the N protein of the MDCK cell 293T correspond to HEK cells bands, as well as the endogenous one. The N2A image has the same problem. Please review and correct the images.

Response 15: We thank the reviewer for pointing this out. We have modified in our revised MS.

Point 16: Why in the supplementary images of the WB there are gels revealed with both antibodies at same membrane and others presented in 2 separate gels? Dis authors strip the membranes?

Response 16: Thanks for your question. WB results for each cell lines were performed on the same membrane by first incubation with antibody against WUXV N protein, after exposure, direct incubation with mouse anti-GAPDH without striping the membranes and then re-exposure. There are two pictures in the row data of HEK-293T and SH-SY5Y  in figure 4, the band of N from the first picture, the band of GAPDH from the second picture.

Point 17: Page 258: Bunyavirus.

Response 17: Thank you for your comments. We have modified in our revised MS as “severe fever with thrombocytopenia syndrome virus”.

Point 18: Page 270: End point after “ORF regions”.

Response 18: We thank the reviewer for pointing this out. We have modified in our revised MS.

Point 19: Page 289: End point after “viral particles”.

Response 19: Thank you for your suggestion. Are your questions aimed at this sentence? “Thirdly, as the main structural element of viral ribonucleoprotein (RNP) and viral particles, the N protein is directly involved in RNA synthesis and viral reproduction.” We still think a comma is more appropriate here.

Point 20: Page 323: End point after “Infection time”.

Response 20: We thank the reviewer for pointing this out. We have modified in our revised MS.

Point 21: Did the authors consider using a Leukocyte cell line to add information to the data presented, given that they also deal with the involvement of the immune system in response to the virus?

Response 21: Thank you for this suggestion. The cell lines selected in our study were considered because of the susceptibility of the virus to different species and different tissue phagocytosis. And also, it is a good scientific comment and it would be interesting to explore it. Although it was not performed here, we will focus on it in the future.

Point 22: Although VERO cells appear resistant and do not form plaques, did the authors verify if the cell was able to replicate the virus through qRT-PCR and/or anti-N-protein WB?

Response 22: Thank you for your question. We performed qRT-PCR to determine the content of viral nucleic acid and detected the expression of viral N protein by WB, the results are negative. All the cells in this study were operated in parallel, and all the results were obtained according to the same operation.

Reviewer 2 Report

The work is interesting since it focuses on studying the infection mechanism of a newly discovered virus but it presents some critical points, mainly the use of  the term ‘Omology’ instead of ‘Similarity’ throughout the text. Let's pay attention to the different meanings of the two terms Omology vs Similatity in biology, considering them synonymously could be a serious mistake. Those sequences that have a common ancestral gene or that share a progenitor are said to be homologous, homology is a qualitative datum. On the contrary the degree of similarity between two sequences can be measured, biological similarity is often due to homology, but it can also occur by chance or due to adaptive convergence phenomena. For this reason i invite you to review the text based on the suggestion above.

Author Response

The work is interesting since it focuses on studying the infection mechanism of a newly discovered virus but it presents some critical points, mainly the use of  the term ‘Omology’ instead of ‘Similarity’ throughout the text. Let's pay attention to the different meanings of the two terms Omology vs Similatity in biology, considering them synonymously could be a serious mistake. Those sequences that have a common ancestral gene or that share a progenitor are said to be homologous, homology is a qualitative datum. On the contrary the degree of similarity between two sequences can be measured, biological similarity is often due to homology, but it can also occur by chance or due to adaptive convergence phenomena. For this reason i invite you to review the text based on the suggestion above.

Response: Thank you for your valuable suggestions for this MS. Your suggestion will make our manuscript more scientific and rigorous. And we have already correct them in our revised MS.

Reviewer 3 Report

In this study, the authors have tested the susceptibility of different cell lines by using cells derived from murine (BHK-21, N2A), human (293T, SH-SY5Y), dog (MDCK), pig (PK-15), monkey (Vero), and chicken (DF1), monitored for monolayer cytopathic effect (CPE). In the present manuscript, they show that BHK-21, MDCK, PK-15, 293T, N2A, SH-SY5Y, and DF1 cells were established to be susceptible to WUXV infection, whereas Vero cells appear to be resistant to infection.

In general, the studies are well done and the manuscript is clearly written.

My comments are shown below, the author may either address these comments or add the limitation in the discussion section.

1.      In the materials and methods section 2.7 of western blotting, the author mentioned that a polyclonal antibody against WUXV N protein (1:1000) was maintained in the author’s laboratory. Could they tell more information about how the antibody was made?

2.      Following the first question, could the polyclonal antibody be suitable for Immunofluorescence (IF). It will be very convictive if the authors can provide some IF data with/without WUXV infection in the cell lines at different time points.

3.      In figure 2, the authors claimed the data have Bars which means two biological replicates, why not in figure 3?

4.      The author shows that Vero cells do not support WUXV replication is very interesting, could the author discuss more about it?

5.      In line 254, the author said the values were means ± SEM, but not shown in Figure 4. Could the author adjust it?

Author Response

Response to reviewer 3 comments

In this study, the authors have tested the susceptibility of different cell lines by using cells derived from murine (BHK-21, N2A), human (293T, SH-SY5Y), dog (MDCK), pig (PK-15), monkey (Vero), and chicken (DF1), monitored for monolayer cytopathic effect (CPE). In the present manuscript, they show that BHK-21, MDCK, PK-15, 293T, N2A, SH-SY5Y, and DF1 cells were established to be susceptible to WUXV infection, whereas Vero cells appear to be resistant to infection.

In general, the studies are well done and the manuscript is clearly written.

My comments are shown below, the author may either address these comments or add the limitation in the discussion section.

We really appreciate you for your rigorous and scientific attitude. Your suggestions are really valuable and helpful for revising and improving our paper. According to your suggestions, we have made the following revisions on this manuscript: 

Point 1: In the materials and methods section 2.7 of western blotting, the author mentioned that a polyclonal antibody against WUXV N protein (1:1000) was maintained in the author’s laboratory. Could they tell more information about how the antibody was made?

Response 1: Thank you for your suggestion. In brief, the N gene of the S segment of WUXV was cloned and expressed in prokaryotic expression. The purified protein was immunized to 6-week-old Balb/c mice four times every 14 days, then the serum was collected to identify the reactivity of polyclonal antibody. WB results showed that the prepared polyclonal antibody could react with the N protein expressed in prokaryotic expression and recognize the viral protein expressed by WUXV infected cells.

Point 2: Following the first question, could the polyclonal antibody be suitable for Immunofluorescence (IF). It will be very convictive if the authors can provide some IF data with/without WUXV infection in the cell lines at different time points.

Response 2: Thank you for your comments. We tried to identify the IF and WB reactivity of the antibody as soon as it was obtained. Unfortunately, the antibody could only detect the viral protein by WB, but not by IF. As we known, in WB, the antibody recognizes the linearized epitope, while in IF, it recognizes the spatial epitope. So, we speculate that the N-protein epitopes recognized by the anti-N antibody were generally inside the protein fold, so when the viral protein is still in the spatial structure, the antibody cannot bind to the corresponding epitope, so it did not show good IF reactivity. And then we're going to make the corresponding antibody again.

Point 3: In figure 2, the authors claimed the data have Bars which means two biological replicates, why not in figure 3?

Response 3: Thank you for your comments. We have plotted each data according to your suggestion in figure 2.

Point 4: The author shows that Vero cells do not support WUXV replication is very interesting, could the author discuss more about it?

Response 4: Thank you for your comments. Vero can support the replication of Rift Valley Fever Virus (RVFV), Toscana virus (TOSV) and Sandfly Fever Sicilian Virus (SFSV), which belong to the same family as WUXV. However, our findings suggest that this cell does not support WUXV replication in three aspects (viral titers, viral nucleic acid content, and N antigen expression). It' s interesting, and it' s one of the directions we' re going to develop next. We have discussed this in our revised MS in line 338-341.

Point 5: In line 254, the author said the values were means ± SEM, but not shown in Figure 4. Could the author adjust it?

Response 5: Thank you for your comments. We are sorry for our careless. For the results in Figure 4, we only performed biological replicates, but not technical replicates. Therefore, there were no means ± SEM for these values. We have corrected in our legend of figure 4.

Round 2

Reviewer 2 Report

I am very glad the authors  reviewed  the paper  in the light of my suggestion based on the use of  the term ‘Omology’ instead of ‘Similarity’; now the manuscript sounds clear and the authors make a systematic  contribution to the research literature in this area of investigation.

Author Response

Dear reviewer

Thank you for your professional advice, your suggestions are really valuable and helpful for revising and improving our paper.

Before submitting our manuscript, we have asked a professional English editing company to complete English editing.
